# Composition Analysis and Nutritional Value Evaluation of Amino Acids in the Fruit of 161 Jujube Cultivars

**DOI:** 10.3390/plants12091744

**Published:** 2023-04-24

**Authors:** Xin Zhao, Bingbing Zhang, Zhi Luo, Ye Yuan, Zhihui Zhao, Mengjun Liu

**Affiliations:** 1College of Horticulture, Hebei Agricultural University, Baoding 071001, China; zhaoxin963741@163.com (X.Z.);; 2Research Center of Chinese Jujube, Hebei Agricultural University, Baoding 071001, China

**Keywords:** jujube, essential amino acids, amino acid ratio coefficient, score of amino acid ratio coefficient, comprehensive score

## Abstract

Amino acids are an essential group of compounds involved in protein synthesis and various metabolic and immune reactions in the human body. Chinese jujubes (*Ziziphus jujuba* Mill.) are an important fruit and medicinal plant which are native to China and have been introduced into around 50 countries. However, systematic research on the composition and content diversity of amino acids in the jujube is still lacking. In this experiment, the amino acid composition and the contents of the dominant amino acids in the fruit of 161 cultivars of jujube were determined by HPLC. Of the twenty-one kinds of amino acids detected, a total of fourteen kinds of amino acids were detected, of which eight kinds of amino acids were relatively high, including five essential amino acids (threonine, valine, isoleucine, leucine, and phenylalanine) and three nonessential amino acids (glycine, alanine, and proline). However, the contents of the remaining six amino acids were relatively low (aspartic acid, glutamic acid, histidine, serine, arginine, and tryptophan). Therefore, the eight primary amino acids were used as the index to evaluate the amino acids of 161 jujube varieties. Proline accounts for 56.8% of the total amino acid content among the eight amino acids. The total content of the eight primary amino acids in most jujube varieties was 1–1.5 g/100 g, and the highest content of ‘Zaoqiangmalianzao’ was 2.356 g/100 g. The average content of proline was 6.01–14.84 times that of the other seven amino acids. According to the WHO/FAO revised model spectrum of ideal essential amino acids for humans, 19 cultivars met the E/T (essential amino acids/total amino acids) standard, and their values ranged from 35% to 45%; 12 cultivars meet E/NE (non-essential amino acids) ≥ 60%. All cultivars reached the requirement of BC (branched–chain amino acids)/E ≥ 40% with 15 cultivars over 68%. One hundred and fifty-seven cultivars reach the standard of BC/A (aromatic amino acids) ≈ 3.0~3.5. The amino acid ratio coefficient analysis showed that phenylalanine was the first limiting amino acid of all the jujube cultivars. The SRC (the score of amino acid ratio coefficient) values of 134 cultivars were between 50% and 70%, with 12 cultivars over 70%, indicating that jujube fruits are of high nutritional value in terms of amino acids. Based on the principal component analysis and comprehensive ranking of amino acid nutritional value, the top five cultivars were screened from the 161 ones tested, i.e., ‘Tengzhouchanghongzao’, ‘Xinzhengxiaoyuanzao’, ‘Hanguowudeng’, ‘Xuputiansuanzao’, and ‘Lichengxiaozao’. This study established, firstly, a complete basic data analysis of amino acid content in jujube fruit which could be used to select germplasm resources suitable for developing functional amino acid food, and provide theoretical support for the high value utilization of amino acids in jujubes.

## 1. Introduction

Chinese jujubes (*Ziziphus jujuba* Mill.), an ancient fruit tree and medicinal plant, have been cultivated for over 3000 years in China [1]. Jujubes are cultivated in over 50 countries but not on a large scale, so more than 95% of the production is still concentrated in China. As one of China’s major fruit cultivars, jujube cultivation covers about 22 million hectares, with more than 90% of the production concentrated in six provinces: Xinjiang, Hebei, Shandong, Shanxi, Shaanxi, and Henan [2]. The annual production of jujubes exceeds nine million tons, ranking first among China’s dried fruits [3]. Modern medical research has shown that jujube fruit can be used for skin care and beauty [4], protecting the liver [5], anti-aging [6], anti-cancer, and boosting immunity [7,8,9]. Furthermore, the fruit is rich in many nutrients, such as amino acids, cyclic nucleotides, sugar, ascorbic acid, and triterpene acid.

Amino acids are a vital group of compounds involved in protein synthesis, metabolism, and the immune response. Additionally, they are critical flavor-active ingredients used as pharmacological components to regulate various physiological activities and even prevent and treat diseases [10,11]. For example, as the primary substrates of glucose synthesis in the liver, alanine, aspartic acid, and glutamic acid play a variety of roles, affecting the immune function of humans and animals. Glycine is an effective antioxidant, which can remove the free radicals required for leukocyte proliferation and antioxidation, reducing the inflammatory reaction and incidence of pathogens in animals [12,13]. Proline is an essential component of collagen and the extracellular matrix. It plays important roles in many physiological functions, such as the regulation of dehydration pressure, redox, and cell proliferation, and modern experience has shown that it is even involved in regulating vascular development [14,15].

Moreover, amino acids can also help fruit trees to form organs and various active substances during the growth process [16]. For example, proline can regulate osmotic balance, prevent water evaporation, and enhance biological stress. Glutamic acid plays a significant role in the photorespiration of nitrogen in plants. L-serine and cysteine are the antagonists of cytokinin, accelerating the senility of leaves.

Although amino acid content is important to assess fruit quality, the research on amino acids in jujube fruits still needs to be improved, as it is especially lacking systematic, rigorous statistical, and comprehensive evaluation. The determination and comparison of amino acids in 161 jujube cultivars were conducted in this work, which would assist in breeding specific varieties with a high abundance of amino acids and provide theoretical support for the development of high-value utilization of amino acids in jujube fruits.

## 2. Results

### 2.1. Amino Acid Composition and Content in Different Cultivars of Jujube Fruits

A high-performance liquid chromatography (HPLC) was employed to determine the contents of 21 kinds of amino acids in jujube fruit. A total of fourteen kinds of amino acids were identified, and the remaining seven ones (methionine, taurine, lysine, cystine, cysteine, asparagine, and glutamine) were not. Among the fourteen amino acids, eight amino acids (threonine, valine, leucine, phenylalanine, and isoleucine, which are essential amino acids, and glycine, proline, and alanine, which are non-essential amino acids) showed higher levels from 0.322 to 1.589 g/100 g in total. Therefore, the eight amino acids with higher content were used as the target amino acids for further tests.

As shown in Table 1, the average TAA (total eight amino acids) content of the 161 jujube cultivars was 1.28 g/100 g, and the coefficient of variation was 1.589. The highest content of TAA was was found in ‘Zaoqiangmalianzao’ (2.356 g/100 g), almost 6.3 times higher than that in sour jujube ‘Xingtai0604’ (0.322 g/100 g). Among the eight kinds of amino acids, proline showed the highest average content (0.727 g/100 g), from 6.01 to 14.84 times that of the other seven kinds of amino acids. Among the 161 jujube cultivars, the highest proline content was also found in ‘Zaoqiangmalianzao’, 1.674 g/100 g, about 15 times that of ‘Puchengzhishezao’. The coefficient of variation of the eight kinds of amino acids ranged from 1.589 to 6.365%.

### 2.2. Nutritional Value Evaluation of Amino Acids in Fruits of Different Jujube Cultivars

According to the ideal amino acid composition proposed by the WHO/FAO: E/T (essential amino acids/total amino acids) ≈ 40% is better for the human body; E/NE (non-essential amino acids) ≥ 60%; BCAA (branched–chain amino acids) should account for 40% of the daily EAA requirements for adults, 41% for children, and 45% for infants; the BC/A (aromatic amino acid) value of normal humans should be 3.0~3.5 [17,18].

EAA is an essential amino acid that cannot be synthesized by the human body and must be taken from food [19]. As can be seen from Figure 1, 19 cultivars meet the E/T standard, and their values range from 35% to 45%. At the same time, 12 cultivars meet the E/N standard, they were ‘Dongadaguazao’, ‘Dalipachizao’, ‘Zaozhuangxiaolingzao’, ‘Zhongningxiaoyuanzao’, ‘Jiaxianbaizao’, Popozao’, ‘Dalizhizao’, ‘Beijingmayabai’, ‘Xinzhengdazao’, ‘Puchengzhishezao’, ‘Zaoqiangcuizao’, and ‘Yucinaitouzao’.

The higher the M/T (medicinal amino acids/total amino acids) ratio, the higher its medicinal value. Figure 1 shows that most M/T ratios were between 15 and 25% for 127 cultivars, and 73 cultivars had M/T ratios higher than 20%. These results implied that jujubes do have rich medicinal amino acids. Flavoring amino acids are those with certain taste sensations involved in forming fruit flavors, thus, causing different flavors in the fruit. The BI/T (bitter amino acids/total amino acids) ratios of 116 jujube cultivars were concentrated between 16 and 28%, accounting for 72.05% of the total number of 161 cultivars. Additionally, all jujube cultivars had SW/T (sweet amino acids/total amino acids) higher than 60%, and 141 had sweet amino acid ratios higher than 72%. The three cultivars with the highest content were ‘Zhongningdanglingzao’ (89.835%), ‘Lengbaiyu’ (89.443%), and ‘Xupumifengzao’ (89.349%). 

Plants contain a large number of free and bound amino acids; bound amino acids are not immediately hydrolyzed when consumed, so free amino acids are more important for taste formation. The taste threshold refers to the minimum concentration of taste components perceived by the taste receptors; the sweet amino acid thresholds are glycine 0.13 g/100 g, threonine 0.26 g/100 g, proline 0.3 g/100 g, and alanine 0.06 g/100 g. The bitter amino acid thresholds are 0.04 g/100 g for valine, 0.09 g/100 g for isoleucine, 0.19 g/100 g for leucine, and 0.09 g/100 g for phenylalanine [20]. Taste Activity Value (TAV) is the ratio of the taste amino acid concentration to its taste threshold, which indicates the contribution of the amino acid to the taste of food. When the TAV ≥ 1, there is a greater effect of the amino acid on the taste characteristics [21]. Eight major amino acids in one hundred and sixty-one date varieties were measured, among which the mean values of the sweet amino acids proline and alanine were above two compared to the taste threshold, while the bitter amino acids were below the taste threshold except for valine (TAV = 1.075), so the sweet amino acids proline and alanine contributed the most to the flavor of jujube fruits. In addition, it was found that when the bitter amino acid content was below its taste threshold, it had the effect of enhancing the freshness and sweetness of other amino acids [22].

BCAAs are not only building blocks of the body’s protein but also regulators of protein, glucose and energy metabolism, and brain function [23]. It could be observed from Figure 1 that all cultivars met the requirement of BC/E ≥ 40%, and even three cultivars of BC/E reached 70%, i.e., ‘Puyangxiaozao’ (76.17), ‘Qufuhoutouzao’ (70.66), and ‘Linzedazao’ (70.13). BC/A is associated with hypertension, and it not only could predict future cardiac events in patients with heart failure but may also influence insulin resistance during pubertal development in girls [24,25,26]. It can be seen from Figure 1 that 157 cultivars reached the BC/A standard. Based on the results, it can be seen that, almost all jujube cultivars met the ideal essential amino acid standard of the human body, and had high nutritional value.

### 2.3. Comparison of RC and SRC of Amino Acids in the Fruit of Jujube Different Cultivars

According to the WHO/FAO scoring model of essential amino acids, RC (the proportionality coefficient of amino acids) = one means the proportion of essential amino acids in food is consistent with the human essential amino acid pattern spectrum. R < one is relatively insufficient, and the amino acid with the lowest content is regarded as the first limiting amino acid. Figure 2 presents that the RC values of valine and isoleucine were close to one, indicating that the contents of these two amino acids in jujube fruit were consistent with the requirements of the human body. We determined the content of the eight primary amino acids in 161 cultivars of jujube fruits, and the results indicated that the highest RC was threonine, near one and a half, showing that the content of threonine was excessive. On the contrary, leucine and phenylalanine were less than one. In comparison, phenylalanine was the first limiting amino acid with an RC of 0–0.767.

Modern research shows that amino acid deficiency affects the nutritional value, so the significance of the SRC (the score of amino acid ratio coefficient) is that if the amino acid composition of the food matches that of the model, the closer the value is to 100, the higher the nutritional value. Figure 3 shows that the SRC value of 134 cultivars was between 50 and 70, accounting for 83.2% of the cultivars tested, and only 12 cultivars had SRC values more than 70%. Among them, the five cultivars with the highest score were ‘Zhongyangtuanzao’ (73.889), ‘Zaoqiangmalianxiaozao’ (72.243), ‘Chaoyangmeixinzao’ (72.181), ‘Leng3’ (71.433) and ‘Hanguowudeng’ (71.318).

### 2.4. Correlation Analysis of the Contents of Amino Acids

We used a comprehensive assessment method to screen the jujube cultivars with high combined amino acid values. Figure 4 provides the correlation between the different amino acid contents. Except for proline and glycine, the correlation of the other six amino acids was higher than 0.7. The correlations among threonine, alanine, valine, isoleucine, leucine, and phenylalanine, as shown in Figure 4, show that the correlation between leucine and the other five amino acids was very high. The highest correlation, between leucine and isoleucine was 0.95, while the correlation between valine and alanine was 0.9, followed by the correlation between leucine and threonine which was as high as 0.83 and 0.87. Except for threonine and isoleucine was 0.76, threonine and phenylalanine were 0.77, and the correlation of these amino acids was above 0.8. However, the correlation between proline and these amino acids was not high (−0.02 to 0.031), nor was it with glycine (−0.02 to 0.41), and even the correlation between proline and glycine was only −0.02. To sum up, threonine, alanine, valine, isoleucine, leucine, and phenylalanine were highly correlated, while proline and glycine differed significantly among the varieties and were independently inherited.

### 2.5. Principal Component Analysis of Amino Acids

Principal component analysis (PCA) is a standard multi-index evaluation method widely used in quantitative trait analysis and comprehensive quality evaluation of crops. The results (Table 2) showed that they mainly reflected the information of principal component one and principal component two. The characteristic values of the first two principal components were more significant than one, with a cumulative contribution rate of 81.867%.

The factor load value reflects the influence of the original variable on the factor. First, positive and negative represent different directions of change. The larger the absolute value of load value, the more significant the influence of amino acids on the principal component. The amino acid content is also high when the principal component is significant. The positive sign indicates a positive effect on the principal component. Conversely, a negative sign indicates a negative effect. According to the principal component analysis, the load coefficients of eight amino acids, threonine, alanine, valine, isoleucine, leucine, and phenylalanine, were highly similar and had a significant contribution to PC 2, ranging from 0.873 to 0.979. On the contrary, proline (0.129) and glycine (0.402) were relatively dispersed. However, proline had an enormous contribution to PC 1 (0.877). On the contrary, glycine had a significant negative correlation with PC 1 (−0.562). The contribution of other amino acids to PC 1 was between −0.081 and 0.16, which was consistent with the Figure 5. Principal component analysis can use load factor information for weight calculation, therefore, this experiment calculated the weight values of 8 amino acids. See Table 3.

### 2.6. Comprehensive Evaluation of Different Cultivars

According to the comprehensive score coefficient calculation, the linear combination coefficient is accumulated after multiplying the variance interpretation rate, respectively, and dividing by the cumulative variance interpretation rate. The weight was calculated, and the weight value of the eight amino acids was obtained by normalizing the comprehensive score coefficient, and construction of the comprehensive evaluation model of the amino acid nutritional value of jujubes.

According to the weight evaluation, we obtained the comprehensive score of 161 jujube cultivars. As shown in the Figure 6, the score of 135 cultivars was concentrated between six and twelve. According to the score from high to low through comprehensive evaluation, the top five cultivars were ‘Tengzhouchanghongzao’ (14.504), ‘Xinzhengxiaoyuanzao’ (13.999), ‘Hanguowudeng’ (13.821), ‘Xuputiansuanzao’ (13.744), and ‘Lichengxiaozao’ (13.708). The fruits from these cultivars are suitable to develop amino acids products.

## 3. Discussion

Amino acids, as an essential nutritional index in jujubes, have been the subject of many relevant studies; in addition, for example, Wang [27] compared sour jujube’s amino acid content levels from six regions. Research has shown that phenylalanine in these different regions of jujubes had a significant negative correlation with single fruit weight and the sugar–acid ratio, and a highly significant positive correlation with total acid and Vc. There was a significant positive correlation between threonine, leucine, and ursolic acid. Wang [28] determined the amino acid content levels of 55 cultivars from eleven counties in Xinjiang. The results showed that the total amount of amino acids in Xinjiang Jun jujubes tested in different production areas was relatively high. The amino acid protein patterns were consistent in similarity. Among them, Xinjiang Jun jujubes from Awati County, Cele County, and Aksu City were more aligned with human protein nutritional needs and had good protein quality evaluations. In addition, the ratio coefficient of the amino acid (RC) of valine in Xinjiang Jun jujubes from different production areas was the lowest, which was the first limiting amino acid, consistent with the results of this experiment. Shi [29] analyzed 20 dried jujube cultivars from seven production areas in North China based on metabolomics. The experimental results showed that there were 54 types of amino acids and their derivatives in dried jujubes, with the content of amino acid derivatives and lipids exceeding 10% in most cultivars. However, there were significant differences between different samples, and dried jujube samples obtained from the same jujube production area had a higher correlation. Therefore, the climate conditions and management models with local characteristics may be the main factors affecting the synthesis and accumulation of metabolic products in jujube fruits. To quantify the nutritional and commercial value of amino acids in different jujube cultivars and to systematically establish the primary amino acid composition and content of different cultivars of jujube fruit, this experiment used high-performance liquid chromatography to determine the content of eight primary amino acids in 161 cultivars. Various evaluation methods were also used for the effective amino acid screening of different jujube cultivars to develop functional foods selectively.

Jujubes are rich in nutrients and often processed into various delicious functional foods such as tea, bread, jujube cakes, jelly, preserves, jams, jujube vinegar, jujube wine [30,31,32], and jujube yogurt [33]. Through the experimental conclusion, we can learn that the level of proline content is high. By adding jujubes to the ingredients, the proline in jujubes can be used as a relevant nutritional supplement to make the product fully functional. Similarly, the SW/T comparison revealed that most jujube cultivars have sweet amino acid ratios above 70%, even ‘Zhongningdanglingzao’, ‘Lengbaiyu’, and ‘Xupumifengzao’ were above 89%, so they can be utilized as raw materials to regulate the flavor of food. The cultivars with high medicinal amino acids/total amino acids ratio, such as ‘Zaozhuangxiaolingzao’ (31.06%), ‘Xinzhengdazao’ (31.24%), and ‘Puchengzhishezao’ (31.69%), can be fully exploited for their medicinal value. In addition, threonine, leucine, and isoleucine were higher in jujubes than the model amino acids. According to the protein complementation method [34], jujubes can be mixed and consumed with other vegetables and fruits to utilize the nutritional value of amino acids efficiently. As a multivariate data processing method, PCA can assess complex and hard-to-find variables for eight significant amino acid differences among 161 jujube cultivars. It can be used for cultivar selection and purposeful cultivation.

It has been shown that in addition to the differences in trace element content in the soil that affects the amino acid content of plants [35], latitude and altitude are also among the main factors affecting the amino acid content [36]. Jujubes are highly adaptable, have outstanding drought tolerance, and can grow well in low fertility. Traditional crops, vegetables, and fruit trees cannot survive in these saline soil environments [37]. The characteristics of jujubes rich in amino acids are likely related to its growth and domestication in extremely harsh environments.

## 4. Materials and Methods

### 4.1. Sample Preparation

To investigate the amino acid composition and content of different cultivars, 161 jujube fruits were selected in this experiment. The fruits of ‘Xingtai0604′ were harvested from Xian County Nursery Base in Hebei Province. The remaining 160 cultivars were harvested from the Shanxi Academy of Agricultural Sciences Polomogy Institute (112°30′36″ E, 37°21′ N); their growth conditions were consistent. After harvesting the denucleated section, the samples were baked at 60 °C to a constant weight, then crushed and passed through a 40-purpose sieve. One gram of each sample was accurately weighed, bagged, and stored in a dry vessel.

### 4.2. Amino Acid Determination

The amino acid determination method draws on the experimental steps of related studies [38]. Pre-treatment of specimens: First, 1.0 g samples were weighed into a glass test tube, and 10 mL of 6 mol/L HCl was added. After thoroughly shaking and mixing, the tubes were capped, and the samples were hydrolyzed for 20 h at 110 °C. After hydrolysis, the samples were cooled to room temperature, taken out of the hydrolysate, centrifuged, and 1.00 mL supernatant was put into a centrifuge tube, and evaporated to dryness using a water bath at 80–90 °C under a stream of nitrogen. The samples were quantitatively transferred into a new centrifuge tube using 2 mL of 0.1 mol/L HCl.

Derivative treatment: The reacting mixture included 200 µL buffer salt solution, 100 μL1-chloro-2,4-dinitrochlorobenzene derivative, 100 µL sample solution, and amino acid standard solution. The derivatization reaction was carried out in a 1.5 mL perforated centrifuge tube. The mixture was then heated at 90 °C for 1.5 h and then cooled to room temperature. An amount of 50 µL of 10% glacial acetic acid solution was added to the tube with 550 µL of purified water and mixed well. Finally, the supernatant in the tube was removed and passed through an organic membrane (0.45 µm) for detection.

Liquid chromatography conditions: Universal C18 column (4.6 mm × 250 mm, 5 m) was used. Ten microliters of the derivatized standard or samples were injected. The solvent system comprised two eluents: flow phase A: sodium acetate–triatomine–glacial acetic acid–pure water (2.5:1.5:1.17:1000, *v*/*v*; flow phase B: 100% pure acetylene; A/B: 82%:18%. Detection was achieved with a UV-vis detector at 360 nm; the column temperature was 40 °C, and the test time was 1 h per sample.

### 4.3. Methods for Evaluating Amino Acids Nutrition

According to WHO/FAO (World Health Organization/Food and Agriculture Organization of the United Nations) revised pattern spectrum of ideal essential amino acids for human body (1973 version), (1) E/T (%) = EAA/TAA, E/N (%) = EAA/NEAA [39], M/T (%) = MAA/TAA. Proportion of flavor amino acids, SW/T (%) = SWAA/TAA, BI/T (%) = BIAA/TAA [40,41], BC/E (%) = BCAA/EAA and BC/A = BCAA/AAA [42]. (2) Ratio of amino acids, RAA, (RAA = EAA of the variety to be evaluated/EAA corresponding to the pattern spectrum). This is used to calculate the multiple between the essential amino acids to be evaluated in the sample and the corresponding amino acids in the WHO/FAO model spectrum. For the RC, the proportionality coefficient of amino acids, RC = average of RAA/RAA, and the amino acid corresponding to the minimum value of RC is the first limiting amino acid [43,44]. The score of amino acid ratio coefficient SRC, SRC = 100 − CV × 100, where CV is the coefficient of variation of RC [45]. (3) Principal component analysis (PCA) was used to select several representative major components by a linear transformation of multiple variables [46,47,48]. Among 161 jujube cultivars, 8 amino acids were found to have high content. Therefore, they were used as the primary amino acids to test and evaluate the nutritional value of amino acids in jujubes to achieve rapid evaluation and screening according to different needs.

### 4.4. Data Statistics and Analysis

Charts were drawn by Excel 2010, and the data in the table were expressed as average values; the correlation and PCA analysis of the contents of various amino acids were performed by SPSSAU (https://spssau.com/index.html (accessed on 15 November 2022). The cumulative variance contribution rate was finally obtained. The higher the value, the better the amino acid quality.

## 5. Conclusions

The TAA content of 62.73% jujube cultivars (101 cultivars) was 1–1.5 g/100 g. ‘Zaoqiang malianzao’ had the highest TAA content of 2.035 g/100 g. The average content of proline was much higher than the other amino acids.

According to the ideal amino acid compositions proposed by WHO/FAO, 8 of 161 jujube cultivars meet the E/T ≥ 40% standard, 12 cultivars meet the standard of E/N ≥ 60%, all cultivars reached the requirement of BC/E ≥ 40%, and even 15 cultivars reached more than 68%. Furthermore, 157 cultivars reached the BC/A ≈ 3.03.5 standards. The M/T ratios of 73 cultivars were higher than 20%. The proportions of sweet amino acid of all cultivars were higher than 60%, and 141 cultivars had a proportion higher than 72%, indicating that amino acids might contribute to the sweet and pleasant taste of jujube fruits.

Phenylalanine was the first limiting amino acid of all jujube cultivars. One hundred and thirty-four out of 161 jujube cultivars had SRC values between 50 and 70, and 12 cultivars had an SRC value over 70%, indicating that jujube fruits have high nutritional value in terms of amino acids.

We obtained the comprehensive ranking of each jujube cultivar. The top five cultivars were ‘Tengzhouchanghongzao’ (14.504), ‘Xinzhengxiaoyuanzao’ (13.999), ‘Hanguowudeng’ (13.821), ‘Xuputiansuanzao’ (13.744), and ‘Lichengxiaozao’ (13.708).

## Figures and Tables

**Figure 1 plants-12-01744-f001:**
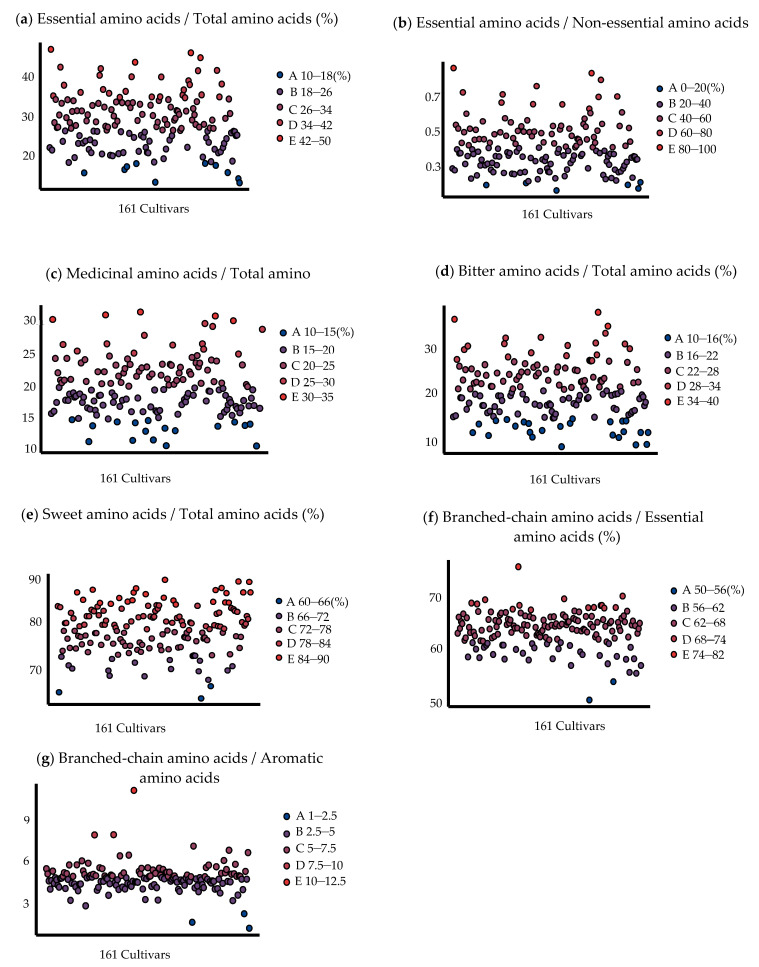
Distribution range of amino acid nutritional value of different jujube cultivars. A–E in turn represents five increasing ranges of values.

**Figure 2 plants-12-01744-f002:**
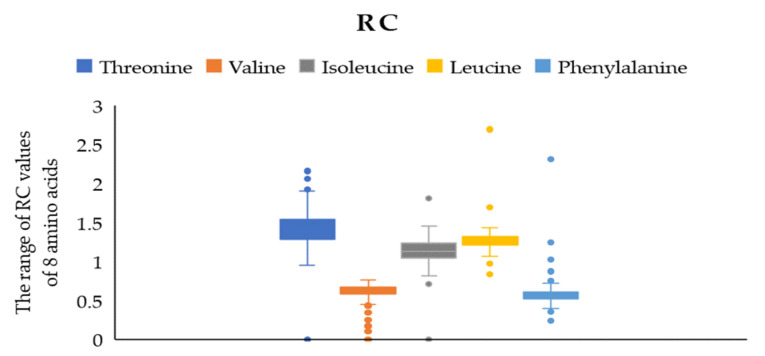
The range of RC values of 5 EAA.

**Figure 3 plants-12-01744-f003:**
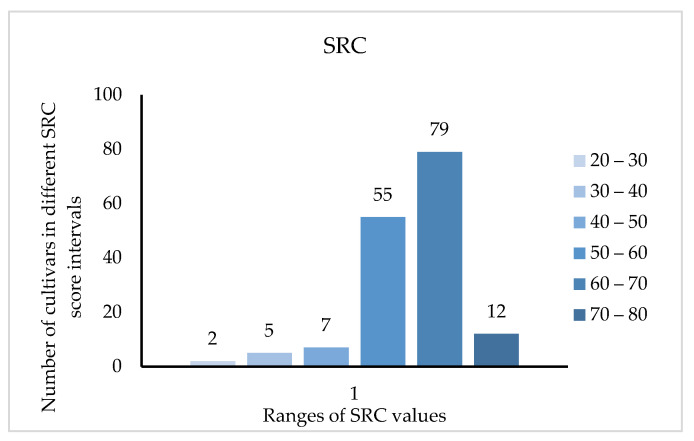
Distribution of SRC value in 161 jujube cultivars.

**Figure 4 plants-12-01744-f004:**
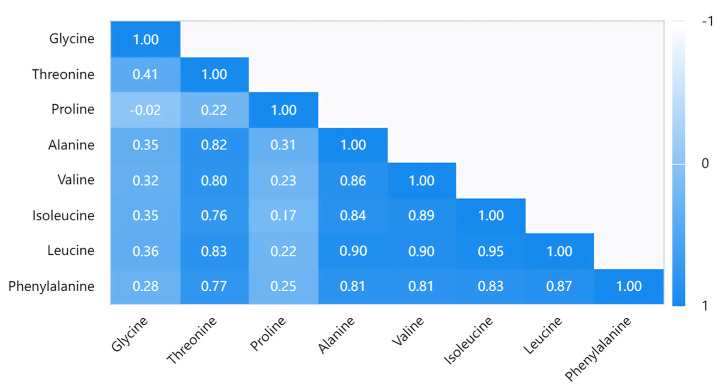
Pearson correlation analysis.

**Figure 5 plants-12-01744-f005:**
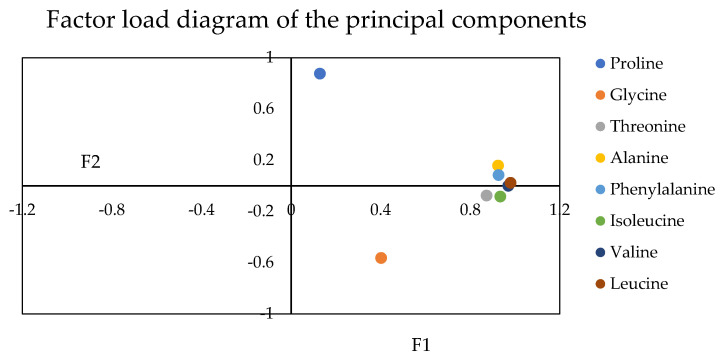
Factor load diagram of the principal components.

**Figure 6 plants-12-01744-f006:**
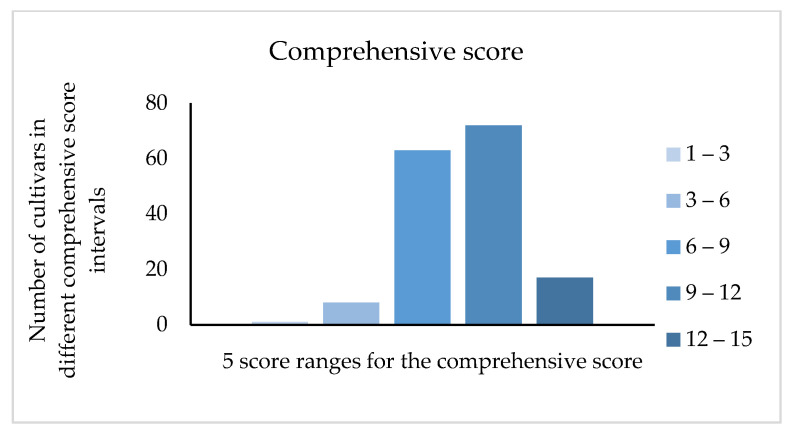
Comprehensive score distribution of 161 cultivars.

**Table 1 plants-12-01744-t001:** Variations of the contents of eight dominant amino acids in the fruits of 161 cultivars.

Amino Acid (AA)	Average (g/100 g)	Coefficient of Variation (%)	The Highest Content (g/100 g)	The Cultivar of Highest Content	The Lowest Content (g/100 g)	The Cultivar of Lowest Content
Glycine	0.079	4.189	0.125	‘Xinzheng dazao’	0.007	‘Xinzheng changjixinzao’
Threonine	0.077	3.680	0.148	‘Jiaxianbaizao’	0.000	‘Xingtai0604′
Proline	0.727	3.009	1.674	‘Zaoqiang malianzao’	0.112	‘Pucheng zhishezao’
Alanine	0.120	6.365	0.176	‘Xinzheng xiaoyuanzao’	0.077	‘Mopanzao’
Valine	0.043	2.596	0.083	‘Dalipachizao’	0.000	‘Xingtai0604’
Isoleucine	0.063	2.866	0.117	‘Dalipachizao’	0.000	‘Xingtai0604’
Leucine	0.121	3.507	0.215	‘Tengzhou changhongzao’	0.024	‘Xingtai0604’
Phenylalanine	0.049	2.449	0.124	‘Linlilajiaozao’	0.010	‘Puchengzhishezao’
Total AA	1.280	1.589	2.356	‘Zaoqiang malianzao’	0.322	‘Xingtai0604’

**Table 2 plants-12-01744-t002:** Total variance interpretations of the principal components analysis.

Principal Components	Initial Eigenvalues	The Percentage of Explained Variance%	Cumulative Variance Contribution%
1	5.418	67.722	67.722
2	1.132	14.145	81.867

**Table 3 plants-12-01744-t003:** Amino acid score coefficient and weight.

Name	F1	F2	Weight (%)
Glycine	0.074	−0.497	6.537102
Threonine	0.161	−0.066	14.22261
Proline	0.024	0.775	2.120141
Alanine	0.17	0.141	15.01767
Valine	0.179	0.001	15.81272
Isoleucine	0.172	−0.072	15.19435
Leucine	0.181	0.023	15.9894
Phenylalanine	0.171	0.076	15.10601

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
