# Peer review of "Composition Analysis and Nutritional Value Evaluation of Amino Acids in the Fruit of 161 Jujube Cultivars"

_plants, 2023, doi:10.3390/plants12091744_

Round 1
Reviewer 1 Report
In the manuscript titled, 'Composition analysis and nutritional value evaluation of amino acids in the fruit of 161 jujube cultivars' the authors present a descriptive study exploring the amino acid content of jujube cultivars. This topic is under reported in the literature. Thus, while I found the novelty of the paper below average, there is a need for this type of report.
Overall, the HPLC methods used were standard and well accepted for amino acid determination. The discussion was brief and to the point. The authors did not overstate any conclusions. The biggest flaw I could find in this study is several instances where spaces are missing between words and numbers or units. Examples can be found on lined 133-135. This could be fixed either by a final editing by the authors, or during the final proofing before publication.
Reviewer 2 Report
Manuscript Plants-2343452 - Composition analysis and nutritional value evaluation of amino acids in the fruit of 161 jujube cultivars
The manuscript is about the amino acid composition and contents of 13 dominant amino acids in the fruit of 161 cultivars of jujube. The relevance of this work concerns on the important role of the amino acids in protein synthesis which are related with various metabolic and immune reactions in the human body. The study reported in this manuscript according to the authors established for the first time a complete basic data of amino acid content in jujube fruit which could be used to 35 select germplasm resources suitable for developing functional amino acid food and provide theoretical support for the high value utilization of amino acids in jujube.
Overall, the manuscript is well done with valuable results for a better production and use of the nutritional and medical properties of jujube. The presentation of the results and the conclusions are consistent with the data obtained during the experiment and are also appropriate for the audience.
However, the manuscript doesn’t have a section with a discussion of the results which, in my opinion should be added. In addition, some suggestions to improve the comprehension of the data could be done:
3. Materials and Methods
1- The authors should refer with some detail the sources of the jujube samples used in this work.
2- Since the study is about the jujube production, the authors can provide some information about the cultivation conditions of jujube trees, the required climatic and edaphic conditions and where (location) this tree is cultivated in China or in other Countries. Also, some information about any published works that related the jujube amino acids composition with its production conditions should also be included (this last approach can be presented in the section “discussion of the results”).
Results
In Fig. 5 the colors and shades used to discriminate the different amino acids are very similar. I suggest using more contrasting colors/shades.
Kind regards
Round 2
Reviewer 2 Report
Authors made the suggested improvements to the manuscript. In my opinion the manuscript can be accepted for publication in the present form.